# BRITS: Bidirectional Recurrent Imputation for Time Series

**Wei Cao**[*]
Tsinghua University
Bytedance AI Lab
cao-13@tsinghua.org.cn

**Dong Wang**
Duke University
dong.wang363@duke.edu

**Jian Li**
Tsinghua University
lijian83@mail.tsinghua.edu.cn

**Hao Zhou**
Bytedance AI Lab
haozhou0806@gmail.com

**Yitan Li**
Bytedance AI Lab
liyitan@bytedance.com

**Lei Li**
Bytedance AI Lab
lileilab@bytedance.com

## Abstract

Time series are ubiquitous in many classification/regression applications. However, the time series data in real applications may contain many missing values. Hence, given multiple (possibly correlated) time series data, it is important to fill in missing values and at the same time to predict their class labels. Existing imputation methods often impose strong assumptions of the underlying data generating process, such as linear dynamics in the state space. In this paper, we propose a novel method, called BRITS, based on recurrent neural networks for missing value imputation in time series data. Our proposed method directly learns the missing values in a bidirectional recurrent dynamical system, without any specific assumption. The imputed values are treated as variables of RNN graph and can be effectively updated during backpropagation. BRITS has three advantages: (a) it can handle multiple correlated missing values in time series; (b) it generalizes to time series with nonlinear dynamics underlying; (c) it provides a data-driven imputation procedure and applies to general settings with missing data. We evaluate our model on three real-world datasets, including an air quality dataset, a health-care dataset, and a localization dataset for human activity. Experiments show that our model outperforms the state-of-the-art methods in both imputation and classification/regression.

## 1 Introduction

Multivariate time series data are abundant in many application areas, such as financial marketing [5, 4], health-care [10, 22], meteorology [31, 26], and traffic engineering [29, 35]. Time series are widely used as signals for classification/regression in various applications of corresponding areas. However, missing values in time series are very common, due to unexpected accidents, such as equipment damage or communication error, and may significantly harm the performance of downstream applications.

Much prior work proposed to fix the missing data problem with statistics and machine learning approaches. However most of them require fairly strong assumptions on missing values. We can fill the missing values using classical statistical time series models such as ARMA or ARIMA (e.g., [1]). But these models are essentially linear (after differencing). Kreindler et al. [19] assume that the data are smoothable, i.e., there is no sudden wave in the periods of missing values, hence imputing missing

---

[*]Work done while Wei Cao was a research intern at Bytedance AI Lab

values can be done by smoothing over nearby values. Matrix completion has also been used to address missing value problems (e.g., [30, 34]). But it typically applies to only static data and requires strong assumptions such as low-rankness. We can also predict missing values by fitting a parametric data-generating model with the observations [14, 2], which assumes that data of time series follow the distribution of hypothetical models. These assumptions make corresponding imputation algorithms less general, and the performance less desirable when the assumptions do not hold.

In this paper, we propose BRITS, a novel method for filling the missing values for multiple correlated time series. Internally, BRITS adapts recurrent neural networks (RNN) [16, 11] for imputing missing values, without any specific assumption over the data. Much prior work uses non-linear dynamical systems for time series prediction [9, 24, 3]. In our method, we instantiate the dynamical system as a bidirectional RNN, i.e., imputing missing values with bidirectional recurrent dynamics. In particular, we make the following technical contributions:

- We design a bidirectional RNN model for imputing missing values. We directly use RNN for predicting missing values, instead of tuning weights for smoothing as in [10]. Our method does not impose specific assumption, hence works more generally than previous methods.

- We regard missing values as variables of the bidirectional RNN graph, which are involved in the backpropagation process. In such case, missing values get delayed gradients in both forward and backward directions with consistency constraints, which makes the estimation of missing values more accurate.

- We simultaneously perform missing value imputation and classification/regression of applications jointly in one neural graph. This alleviates the error propagation problem from imputation to classification/regression and makes the classification/regression more accurate.

- We evaluate our model on three real-world datasets, including an air quality dataset, a health-care dataset and a localization dataset of human activities. Experimental results show that our model outperforms the state-of-the-art models for both imputation and classification/regression accuracies.

## 2    Related Work

There is a large body of literature on the imputation of missing values in time series. We only mention a few closely related ones. The *interpolate method* tries to fit a "smooth curve" to the observations and thus reconstruct the missing values by the local interpolations [19, 14]. Such method discards any relationships between the variables over time. The *autoregressive method*, including ARIMA, SARIMA etc., eliminates the non-stationary parts in the time series data and fit a parameterized stationary model. The *state space model* further combines ARIMA and Kalman Filter [14, 15], which provides more accurate results. *Multivariate Imputation by Chained Equations* (MICE) [2] first initializes the missing values arbitrarily and then estimates each missing variable based on the chain equations. The graphical model [20] introduces a latent variable for each missing value, and finds the latent variables by learning their transition matrix. There are also some data-driven methods for missing value imputation. Yi et al. [32] imputed the missing values in air quality data with geographical features. Wang et al. [30] imputed missing values in recommendation system with collaborative filtering. Yu et al. [34] utilized matrix factorization with temporal regularization to impute the missing values in regularly sampled time series data.

Recently, some researchers attempted to impute the missing values with recurrent neural networks [7, 10, 21, 12, 33]. The recurrent components are trained together with the classification/regression component, which significantly boosts the accuracy. Che et al. [10] proposed GRU-D, which imputes missing values in health-care data with a smooth fashion. It assumes that a missing variable can be represented as the combination of its corresponding last observed value and the global mean. GRU-D achieves remarkable performance on health-care data. However, it has many limitations on general datasets [10]. A closely related work is M-RNN proposed by Yoon et al. [33]. M-RNN also utilizes bi-directional RNN to impute missing values. Differing from our model, it drops the relationships between missing variables. The imputed values in M-RNN are treated as constants and cannot be sufficiently updated.

## 3  Preliminary

We first present the problem formulation and some necessary preliminaries.

**Definition 1 (Multivariate Time Series)** *We denote a multivariate time series* $\mathbf{X} = \{\mathbf{x}_1, \mathbf{x}_2, \ldots, \mathbf{x}_T\}$ *as a sequence of* $T$ *observations. The* $t$-*th observation* $\mathbf{x}_t \in \mathbb{R}^D$ *consists of* $D$ *features* $\{x_t^1, x_t^2, \ldots, x_t^D\}$, *and was observed at timestamp* $s_t$ *(the time gap between different timestamps may not be the same). In reality, due to unexpected accidents, such as equipment damage or communication error,* $\mathbf{x}_t$ *may have the missing values (e.g., in Fig. 1,* $x_1^3$ *in* $\mathbf{x}_1$ *is missing). To represent the missing values in* $\mathbf{x}_t$, *we introduce a masking vector* $\mathbf{m}_t$ *where,*

$$\mathbf{m}_t^d = \begin{cases} 0 & \text{if } x_t^d \text{ is not observed} \\ 1 & \text{otherwise} \end{cases} .$$

*In many cases, some features can be missing for consecutive timestamps (e.g., blue blocks in Fig. 1). We define* $\delta_t^d$ *to be the time gap from the last observation to the current timestamp* $s_t$, *i.e.,*

$$\delta_t^d = \begin{cases} s_t - s_{t-1} + \delta_{t-1}^d & \text{if } t > 1, \mathbf{m}_{t-1}^d = 0 \\ s_t - s_{t-1} & \text{if } t > 1, \mathbf{m}_{t-1}^d = 1 \\ 0 & \text{if } t = 1 \end{cases} .$$

*See Fig. 1 for an illustration.*

| time series X | | | | | | masking vectors | | | | | | time gaps | | | | | | |
|---|---|---|---|---|---|---|---|---|---|---|---|---|---|---|---|---|---|---|
| 31 | / | / | 32 | 27 | 22 | 1 | 0 | 0 | 1 | 1 | 1 | 0 | 2 | 7 | 9 | 5 | 1 | $d=1$ |
| 6 | 17 | / | / | / | 13 | 1 | 1 | 0 | 0 | 0 | 1 | 0 | 2 | 5 | 7 | 12 | 13 | $d=2$ |
| / | 107 | / | 87 | 66 | 90 | 0 | 1 | 0 | 1 | 1 | 1 | 0 | 2 | 5 | 7 | 5 | 1 | $d=3$ |
| $\mathbf{x}_1$ | $\mathbf{x}_2$ | ...... | | | $\mathbf{x}_6$ | $\mathbf{m}_1$ | $\mathbf{m}_2$ | | ...... | | $\mathbf{m}_6$ | $\delta_1$ | $\delta_2$ | | ...... | | $\delta_6$ | |

Figure 1: An example of multivariate time series with missing values. $\mathbf{x}_1$ to $\mathbf{x}_6$ are observed at $s_{1\ldots6} = 0, 2, 7, 9, 14, 15$ respectively. Considering the 2nd feature in $\mathbf{x}_6$, the last observation of the 2nd feature took place at $s_2 = 2$, and we have that $\delta_6^2 = s_6 - s_2 = 13$.

In this paper, we study a general setting for time series classification/regression problems with missing values. We use $\mathbf{y}$ to denote the label of corresponding classification/regression task. In general, $\mathbf{y}$ can be either a scalar or a vector. Our goal is to predict $\mathbf{y}$ based on the given time series $\mathbf{X}$. In the meantime, we impute the missing values in $\mathbf{X}$ as accurate as possible. In another word, we aim to design an effective multi-task learning algorithm for both classification/regression and imputation.

## 4  BRITS

In this section, we describe the BRITS. Differing from the prior work which uses RNN to impute missing values in a smooth fashion [10], we learn the missing values directly in a recurrent dynamical system [25, 28] based on the observed data. The missing values are thus imputed according to the recurrent dynamics, which significantly boosts both the imputation accuracy and the final classification/regression accuracy. We start the description with the simplest case: the variables observed at the same time are mutually uncorrelated. For such case, we propose algorithms for imputation with *unidirectional recurrent dynamics* and *bidirectional recurrent dynamics*, respectively. We further propose an algorithm for correlated multivariate time series in Section 4.3.

### 4.1  Unidirectional Uncorrelated Recurrent Imputation

For the simplest case, we assume that for the $t$-th step, $x_t^i$ and $x_t^j$ are uncorrelated if $i \neq j$ (but $x_t^i$ may be correlated with some $x_{t' \neq t}^j$). We first propose an imputation algorithm by unidirectional recurrent dynamics, denoted as **RITS-I**.

In a *unidirectional recurrent dynamical system*, each value in the time series can be derived by its predecessors with a fixed *arbitrary* function [9, 24, 3]. Thus, we iteratively impute all the variables in

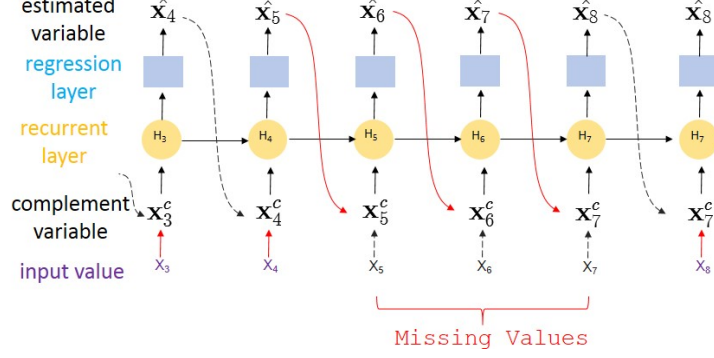

Figure 2: Imputation with unidirectional dynamics.

the time series according to the recurrent dynamics. For the $t$-th step, if $\mathbf{x}_t$ is actually observed, we use it to validate our imputation and pass $\mathbf{x}_t$ to the next recurrent steps. Otherwise, since the future observations are correlated with the current value, we replace $\mathbf{x}_t$ with the obtained imputation, and validate it by the future observations. To be more concrete, let us consider an example.

**Example 1** *Suppose we are given a time series* $\mathbf{X} = \{\mathbf{x}_1, \mathbf{x}_2, \ldots, \mathbf{x}_{10}\}$, *where* $\mathbf{x}_5, \mathbf{x}_6$ *and* $\mathbf{x}_7$ *are missing[2]. According to the recurrent dynamics, at each time step* $t$, *we can obtain an estimation* $\hat{\mathbf{x}}_t$ *based on the previous* $t-1$ *steps. In the first* $4$ *steps, the estimation error can be obtained immediately by calculating the estimation loss function* $\mathcal{L}_e(\hat{\mathbf{x}}_t, \mathbf{x}_t)$ *for* $t = 1, \ldots, 4$. *However, when* $t = 5, 6, 7$, *we cannot get the error immediately since the true values are missing. Nevertheless, note that at the* $8$-*th step,* $\hat{\mathbf{x}}_8$ *depends on* $\hat{\mathbf{x}}_5$ *to* $\hat{\mathbf{x}}_7$. *We thus obtain a "delayed" error for* $\hat{\mathbf{x}}_{t=5,6,7}$ *at the* $8$-*th step.*

### 4.1.1 Algorithm

We introduce a recurrent component and a regression component for imputation. The recurrent component is achieved by a recurrent neural network and the regression component is achieved by a fully-connected network. A standard recurrent network [17] can be represented as

$$\mathbf{h}_t = \sigma(\mathbf{W}_h \mathbf{h}_{t-1} + \mathbf{U}_h \mathbf{x}_t + \mathbf{b}_h),$$

where $\sigma$ is the sigmoid function, $\mathbf{W}_h$, $\mathbf{U}_h$ and $\mathbf{b}_h$ are parameters, and $\mathbf{h}_t$ is the hidden state of previous time steps.

In our case, since $\mathbf{x}_t$ may have missing values, we cannot use $\mathbf{x}_t$ as the input directly as in the above equation. Instead, we use a "complement" input $\mathbf{x}_t^c$ derived by our algorithm when $\mathbf{x}_t$ is missing. Formally, we initialize the initial hidden state $\mathbf{h}_0$ as an all-zero vector and then update the model by:

$$\hat{\mathbf{x}}_t = \mathbf{W}_x \mathbf{h}_{t-1} + \mathbf{b}_x, \tag{1}$$

$$\mathbf{x}_t^c = \mathbf{m}_t \odot \mathbf{x}_t + (1 - \mathbf{m}_t) \odot \hat{\mathbf{x}}_t, \tag{2}$$

$$\gamma_t = \exp\{-\max(0, \mathbf{W}_\gamma \delta_t + \mathbf{b}_\gamma)\}, \tag{3}$$

$$\mathbf{h}_t = \sigma(\mathbf{W}_h[\mathbf{h}_{t-1} \odot \gamma_t] + \mathbf{U}_h[\mathbf{x}_t^c \circ \mathbf{m}_t] + \mathbf{b}_h), \tag{4}$$

$$\ell_t = \langle \mathbf{m}_t, \mathcal{L}_e(\mathbf{x}_t, \hat{\mathbf{x}}_t) \rangle. \tag{5}$$

Eq. (1) is the regression component which transfers the hidden state $\mathbf{h}_{t-1}$ to the estimated vector $\hat{\mathbf{x}}_t$. In Eq. (2), we replace missing values in $\mathbf{x}_t$ with corresponding values in $\hat{\mathbf{x}}_t$, and obtain the complement vector $\mathbf{x}_t^c$. Besides, since the time series may be irregularly sampled, in Eq. (3), we further introduce a *temporal decay factor* $\gamma_t$. Such factor represents the missing patterns in the time series which is critical to imputation [10]. In Eq. (4), based on the decayed hidden state, we predict the next state $\mathbf{h}_t$. Here, $\circ$ indicates the concatenate operation. In the mean time, we calculate the estimation error by the estimation loss function $\mathcal{L}_e$ in Eq. (5). In our experiment, we use the *mean absolute error* for $\mathcal{L}_e$. Finally, we predict the task label $\mathbf{y}$ as

$$\hat{\mathbf{y}} = f_{out}(\sum_{i=1}^{T} \alpha_i \mathbf{h}_i),$$

where $f_{out}$ can be either a fully-connected layer or a softmax layer depending on the specific task, and $\alpha_i$ is the weight for different hidden state which can be derived by the *attention mechanism* or the *mean pooling mechanism*[3], i.e., $\alpha_i = \frac{1}{T}$. We calculate the output loss by $\mathcal{L}_{out}(\mathbf{y}, \hat{\mathbf{y}})$. Our model is then updated by minimizing the accumulated loss $\frac{1}{T}\sum_{t=1}^{T}\ell_t + \mathcal{L}_{out}(\mathbf{y}, \hat{\mathbf{y}})$.

### 4.1.2 Practical Issues

In practice, we use LSTM as the recurrent component in Eq. (4) to prevent the gradient vanishing/exploding problems in vanilla RNN [17]. Standard RNN models including LSTM treat $\hat{\mathbf{x}}_t$ as a constant. During backpropagation, gradients are cut at $\hat{\mathbf{x}}_t$. This makes the estimation errors backpropagate insufficiently. For example, in Example 1, the estimation errors of $\hat{\mathbf{x}}_5$ to $\hat{\mathbf{x}}_7$ are obtained at the 8-th step as the delayed errors. However, if we treat $\hat{\mathbf{x}}_5$ to $\hat{\mathbf{x}}_7$ as constants, such delayed error cannot be fully backpropagated. To overcome such issue, we treat $\hat{\mathbf{x}}_t$ as a variable of RNN graph. We let the estimation error passes through $\hat{\mathbf{x}}_t$ during the backpropagation. Fig. 2 shows how RITS-I method works in Example 1. In this example, the gradients are backpropagated through the opposite direction of solid lines. Thus, the delayed error $\ell_8$ is passed to steps $5, 6$ and $7$. In the experiment, we find that our models are unstable during training if we treat $\hat{\mathbf{x}}_t$ as a constant. See the appendix for details.

### 4.2 Bidirectional Uncorrelated Recurrent Imputation

In the RITS-I, errors of estimated missing values are delayed until the presence of the next observation. For example, in Example 1, the error of $\hat{\mathbf{x}}_5$ is delayed until $\mathbf{x}_8$ is observed. Such error delay makes the model converge slowly and in turn leads to inefficiency in training. In the meantime, it also leads to the *bias exploding* problem [6], i.e., the mistakes made early in the sequential prediction are fed as input to the model and may be quickly amplified. In this section, we propose an improved version called **BRITS-I**. The algorithm alleviates the above-mentioned issues by utilizing the *bidirectional recurrent dynamics* on the given time series, i.e., besides the forward direction, each value in time series can be also derived from the backward direction by another fixed arbitrary function.

To illustrate the intuition of BRITS-I algorithm, again, we consider Example 1. Consider the backward direction of the time series. In bidirectional recurrent dynamics, the estimation $\hat{\mathbf{x}}_4$ reversely depends on $\hat{\mathbf{x}}_5$ to $\hat{\mathbf{x}}_7$. Thus, the error in the 5-th step is back-propagated from not only the 8-th step in the forward direction (which is far from the current position), but also the 4-th step in the backward direction (which is closer). Formally, the BRITS-I algorithm performs the RITS-I as shown in Eq. (1) to Eq. (5) in forward and backward directions, respectively. In the forward direction, we obtain the estimation sequence $\{\hat{\mathbf{x}}_1, \hat{\mathbf{x}}_2, \ldots, \hat{\mathbf{x}}_T\}$ and the loss sequence $\{\ell_1, \ell_2, \ldots, \ell_T\}$. Similarly, in the backward direction, we obtain another estimation sequence $\{\hat{\mathbf{x}}'_1, \hat{\mathbf{x}}'_2, \ldots, \hat{\mathbf{x}}'_T\}$ and another loss sequence $\{\ell_1', \ell_2', \ldots, \ell_T'\}$. We enforce the prediction in each step to be consistent in both directions by introducing the "*consistency loss*":

$$\ell_t^{cons} = \text{Discrepancy}(\hat{\mathbf{x}}_t, \hat{\mathbf{x}}'_t) \tag{6}$$

where we also use the mean absolute error as the discrepancy in our experiment. The final estimation loss is obtained by accumulating the forward loss $\ell_t$, the backward loss $\ell'_t$ and the consistency loss $\ell_t^{cons}$. The final estimation in the $t$-th step is the mean of $\hat{\mathbf{x}}_t$ and $\hat{\mathbf{x}}_t'$.

### 4.3 Correlated Recurrent Imputation

The previously proposed algorithms RITS-I and BRITS-I assume that features observed at the same time are mutually uncorrelated. This may be not true in some scenarios. For example, in the air quality data [32], each feature represents one measurement in a monitoring station. Obviously, the observed measurements are spatially correlated. In general, the measurement in one monitoring station is close to those values observed in its neighboring stations. In this case, we can estimate a missing measurement according to both its historical data, and the measurements in its neighbors.

In this section, we propose an algorithm, which utilizes the feature correlations in the unidirectional recurrent dynamical system. We refer to such algorithm as **RITS**. The feature correlated algorithm for bidirectional case (i.e., **BRITS**) can be derived in the same way. Note that in Section 4.1, the estimation $\hat{\mathbf{x}}_t$ is only correlated with the historical observations, but irrelevant with the the current

observation. We refer to $\hat{\mathbf{x}}_t$ as a "history-based estimation". In this section, we derive another "feature-based estimation" for each $x_t^d$, based on the other features at time $s_t$. Specifically, at the $t$-th step, we first obtain the complement observation $\mathbf{x}_t^c$ by Eq. (1) and Eq. (2). Then, we define our feature-based estimation as $\hat{\mathbf{z}}_t$ where

$$\hat{\mathbf{z}}_t = \mathbf{W}_z \mathbf{x}_t^c + \mathbf{b}_z, \tag{7}$$

$\mathbf{W}_z$, $\mathbf{b}_z$ are corresponding parameters. We restrict the diagonal of parameter matrix $\mathbf{W}_z$ to be all zeros. Thus, the $d$-th element in $\hat{\mathbf{z}}_t$ is exactly the estimation of $x_t^d$, based on the other features. We further combine the historical-based estimation $\hat{\mathbf{x}}_t$ and the feature-based estimation $\hat{\mathbf{z}}_t$. We denote the combined vector as $\hat{\mathbf{c}}_t$, and we have that

$$\begin{aligned} \beta_t &= \sigma(\mathbf{W}_\beta[\gamma_t \circ \mathbf{m}_t] + \mathbf{b}_\beta) \tag{8} \\ \hat{\mathbf{c}}_t &= \beta_{\mathbf{t}} \odot \hat{\mathbf{z}}_t + (1 - \beta_{\mathbf{t}}) \odot \hat{\mathbf{x}}_t. \tag{9} \end{aligned}$$

Here we use $\beta_t \in [0,1]^D$ as the weight of combining the history-based estimation $\hat{\mathbf{x}}_t$ and the feature-based estimation $\hat{\mathbf{z}}_t$. Note that $\hat{\mathbf{z}}_t$ is derived from $\mathbf{x}_t^c$ by Eq. (7). The elements of $\mathbf{x}_t^c$ can be history-based estimations or truly observed values, depending on the presence of the observations. Thus, we learn the weight $\beta_t$ by considering both the temporal decay $\gamma_t$ and the masking vector $\mathbf{m}_t$ as shown in Eq. (8). The rest parts are similar as the feature uncorrelated case. We first replace missing values in $\mathbf{x}_t$ with the corresponding values in $\hat{\mathbf{c}}_t$. The obtained vector is then fed to the next recurrent step to predict memory $\mathbf{h}_t$:

$$\begin{aligned} \mathbf{c}_t^c &= \mathbf{m}_t \odot \mathbf{x}_t + (1 - \mathbf{m}_t) \odot \hat{\mathbf{c}}_t \tag{10} \\ \mathbf{h}_t &= \sigma(\mathbf{W}_h[\mathbf{h}_{t-1} \odot \gamma_t] + \mathbf{U}_h[\mathbf{c}_t^c \circ \mathbf{m}_t] + \mathbf{b}_h). \tag{11} \end{aligned}$$

However, the estimation loss is slightly different with the feature uncorrelated case. We find that simply using $\ell_t = \mathcal{L}_e(\mathbf{x}_t, \hat{\mathbf{c}}_t)$ leads to a very slow convergence speed. Instead, we accumulate all the estimation errors of $\hat{\mathbf{x}}_t$, $\hat{\mathbf{z}}_t$ and $\hat{\mathbf{c}}_t$:

$$\ell_t = \mathcal{L}_e(\mathbf{x}_t, \hat{\mathbf{x}}_t) + \mathcal{L}_e(\mathbf{x}_t, \hat{\mathbf{z}}_t) + \mathcal{L}_e(\mathbf{x}_t, \hat{\mathbf{c}}_t).$$

## 5 Experiment

Our proposed methods are applicable to a wide variety of applications. We evaluate our methods on three different real-world datasets. The download links of the datasets, as well as the implementation codes can be found in the GitHub page[4].

### 5.1 Dataset Description

#### 5.1.1 Air Quality Data

We evaluate our models on the air quality dataset, which consists of PM2.5 measurements from 36 monitoring stations in Beijing. The measurements are hourly collected from 2014/05/01 to 2015/04/30. Overall, there are 13.3% values are missing. For this dataset, we do pure imputation task. We use exactly the same train/test setting as in prior work [32], i.e., we use the 3rd, 6th, 9th and 12th months as the test data and the other months as the training data. See the appendix for details. To train our model, we randomly select every 36 consecutive steps as one time series.

#### 5.1.2 Health-care Data

We evaluate our models on health-care data in *PhysioNet Challenge 2012* [27], which consists of 4000 multivariate clinical time series from intensive care unit (ICU). Each time series contains 35 measurements such as Albumin, heart-rate etc., which are irregularly sampled at the first 48 hours after the patient's admission to ICU. We stress that this dataset is extremely sparse. There are up to 78% missing values in total. For this dataset, we do both the imputation task and the classification task. To evaluate the imputation performance, we randomly eliminate 10% of observed measurements from data and use them as the ground-truth. At the same time, we predict the in-hospital death of each patient as a binary classification task. Note that the eliminated measurements are only used for validating the imputation, and they are never visible to the model.

### 5.1.3 Localization for Human Activity Data

The UCI localization data for human activity [18] contains records of five people performing different activities such as walking, falling, sitting down etc (there are 11 activities). Each person wore four sensors on her/his left/right ankle, chest, and belt. Each sensor recorded a 3-dimensional coordinates for about 20 to 40 millisecond. We randomly select 40 consecutive steps as one time series, and there are 30, 917 time series in total. For this dataset, we do both imputation and classification tasks. Similarly, we randomly eliminate 10% observed data as the imputation ground-truth. We further predict the corresponding activity of observed time series (i.e., walking, sitting, etc).

## 5.2 Experiment Setting

### 5.2.1 Model Implementations

To make a fair comparison, we control the number of parameters of all models as around 80, 000. We train our model by an Adam optimizer with learning rate 0.001 and batch size 64. For all the tasks, we normalize the numerical values to have zero mean and unit variance for stable training.

We use different early stopping strategies for pure imputation task and classification tasks. For the imputation tasks, we randomly select 10% of non-missing values as the validation data. The early stopping is thus performed based on the validation error. For the classification tasks, we first pre-train the model as a pure imputation task and report its imputation accuracy. Then we use 5-fold cross validation to further optimize both the imputation and classification losses simultaneously.

We evaluate the imputation performance in terms of *mean absolute error* (MAE) and *mean relative error* (MRE). Suppose that $\texttt{label}_i$ is the ground-truth of the $i$-th item, $\texttt{pred}_i$ is the output of the $i$-th item, and there are $N$ items in total. Then, MAE and MRE are defined as

$$\texttt{MAE} = \frac{\sum_i |\texttt{pred}_i - \texttt{label}_i|}{N}, \quad \texttt{MRE} = \frac{\sum_i |\texttt{pred}_i - \texttt{label}_i|}{\sum_i |\texttt{label}_i|}.$$

For air quality data, the evaluation is performed on its original data. For heath-care data and activity data, since the numerical values are not in the same scale, we evaluate the performances on their normalized data. To further evaluate the classification performances, we use *area under ROC curve* (AUC) [8] for health-care data, since such data is highly unbalanced (there are 10% patients who died in hospital). We then use standard accuracy for the activity data, since different activities are relatively balanced.

### 5.2.2 Baseline Methods

We compare our model with both RNN-based methods and non-RNN based methods. The non-RNN based imputation methods include:

- **Mean:** We simply replace the missing values with corresponding global mean.

- **KNN:** We use $k$-nearest neighbor [13] (with normalized Euclidean distance) to find the similar samples, and impute the missing values with weighted average of its neighbors.

- **Matrix Factorization (MF):** We factorize the data matrix into two low-rank matrices, and fill the missing values by matrix completion [13].

- **MICE**: We use Multiple Imputation by Chained Equations (MICE), a widely used imputation method, to fill the missing values. MICE creates multiple imputations with chained equations [2].

- **ImputeTS**: We use ImputeTS package in R to impute the missing values. ImputeTS is a widely used package for missing value imputation, which utilizes the state space model and kalman smoothing [23].

- **STMVL**: Specifically, we use STMVL for the air quality data imputation. STMVL is the state-of-the-art method for air quality data imputation. It further utilizes the geo-locations when imputing missing values [32].

Table 1: Performance Comparison for Imputation Tasks (in MAE(MRE%))

| Method | | Air Quality | Health-care | Human Activity |
|---|---|---|---|---|
| Non-RNN | Mean | 55.51 (77.97%) | 0.461 (65.61%) | 0.767 (96.43%) |
| | KNN | 29.79 (41.85%) | 0.367 (52.15%) | 0.479 (58.54%) |
| | MF | 27.94 (39.25%) | 0.468 (67.97%) | 0.879 (110.44%) |
| | MICE | 27.42 (38.52%) | 0.510 (72.5%) | 0.477 (57.94%) |
| | ImputeTS | 19.58 (27.51%) | 0.390 (54.2%) | 0.363 (45.65%) |
| | STMVL | 12.12 (17.40%) | / | / |
| RNN | GRU-D | / | 0.559 (77.58%) | 0.558 (70.05%) |
| | M-RNN | 14.05 (20.16%) | 0.445 (61.87%) | 0.248 (31.19%) |
| Ours | RITS-I | 12.45 (17.93%) | 0.385 (53.41%) | 0.240 (30.10%) |
| | BRITS-I | 11.58 (16.66%) | 0.361 (50.01%) | 0.220 (27.61%) |
| | RITS | 12.19 (17.54%) | 0.292 (40.82%) | 0.248 (31.21%) |
| | **BRITS** | **11.56 (16.65%)** | **0.278 (38.72%)** | **0.219 (27.59%)** |

We implement KNN, MF and MICE based on the python package fancyimpute[5]. In recent studies, RNN-based models in missing value imputations achieve remarkable performances [10, 21, 12, 33]. We also compare our model with existing RNN-based imputation methods, including:

- **GRU-D:** GRU-D is proposed for handling missing values in health-care data. It imputes each missing value by the weighted combination of its last observation, and the global mean, together with a recurrent component [10].

- **M-RNN:** M-RNN also uses bi-directional RNN. It imputes the missing values according to hidden states in both directions in RNN. M-RNN treats the imputed values as constants. It does not consider the correlations among different missing values [33].

We compare the baseline methods with our four models: RITS-I (Section 4.1), RITS (Section 4.2), BRITS-I (Section 4.3) and BRITS (Section 4.3). We implement all the RNN-based models with PyTorch, a widely used package for deep learning. All models are trained with GPU GTX 1080.

## 5.3 Experiment Results

Table 1 shows the imputation results. As we can see, simply applying naïve mean imputation is very inaccurate. Alternatively, KNN, MF, and MICE perform much better than mean imputation. However, these methods demonstrate unstable performances in different tasks. For example, the MF algorithm performs well on the health-care data. But it shows a very bad accuracy on the human activity data. ImputeTS achieves the best performance among all the non-RNN methods, especially for the heath-care data (which is smooth and contains few sudden waves). STMVL performs well on the air quality data. However, it is specifically designed for geographical data, and cannot be applied in the other datasets. Most of RNN-based methods, except GRU-D, demonstrates significantly better performances in imputation tasks. We stress that GRU-D imputes missing value implicitly. It actually performs very well on classification accuracies. M-RNN uses explicitly imputation procedure, and achieves remarkable imputation results. Our model BRITS outperforms all the baseline models. According to the performances of our four models, we also find that both bidirectional recurrent dynamics, and the feature correlations are helpful to enhance the model performance.

We further compare classification accuracies as shown in Table 2. Similar as the imputation tasks, our model BRITS outperforms all the other RNN-based models in classification tasks as well. Note that despite GRU-D does not demonstrate accurate imputation results, it actually performs very well on the classification tasks. The AUC score of GRU-D is only slightly worse than our RITS model. To further show the correlations between imputation accuracy and classification accuracy, we do the health-care classification based on the imputed values by different models, with the classical random forest algorithm. The results are shown in Fig. 3. Surprisingly, we find that the random forest

actually works well with simple mean imputation. The AUC score based on the mean imputation is even better than that on GRU-D and M-RNN. We guess that since GRU-D and M-RNN does not focus on the imputation accuracy, the imputed values might be harmful to downstream classifications with the other models. Alternatively, our model BRITS uses a multi-task learning mechanism which effectively enhances the classification accuracy.

Table 2: Performance Comparison for Classification Tasks

| Method | Health-care (AUC) | Human Activity (Accuracy) |
|---|---|---|
| GRU-D | $0.834 \pm 0.002$ | $0.940 \pm 0.010$ |
| M-RNN | $0.817 \pm 0.003$ | $0.938 \pm 0.010$ |
| RITS-I | $0.821 \pm 0.007$ | $0.934 \pm 0.008$ |
| BRITS-I | $0.831 \pm 0.003$ | $0.940 \pm 0.012$ |
| RITS | $0.840 \pm 0.004$ | $0.968 \pm 0.010$ |
| **BRITS** | $\mathbf{0.850 \pm 0.002}$ | $\mathbf{0.969 \pm 0.008}$ |

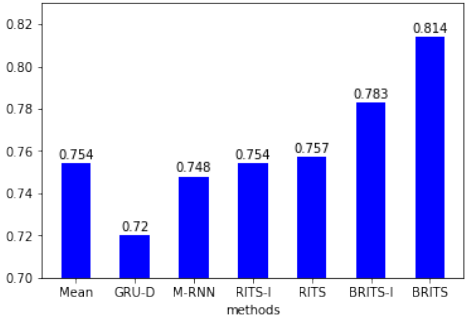

Figure 3: Health-care Classification Based on Different Imputations with Random Forest

# 6   Conclusion

In this paper, we proposed BRITS, a novel method to use recurrent dynamics to effectively impute the missing values in multivariate time series. Instead of imposing assumptions over the data-generating process, our model directly learns the missing values in a bidirectional recurrent dynamical system, without any specific assumption. Our model treats missing values as variables of the bidirectional RNN graph. Thus, we get the delayed gradients for missing values in both forward and backward directions, which makes the imputation of missing values more accurate. We performed the missing value imputation and classification/regression simultaneously within a joint neural network. Experiment results show that our model demonstrates more accurate results for both imputation and classification/regression than state-of-the-art methods.

# 7   Acknowledgements

Wei Cao and Jian Li are supported in part by the National Basic Research Program of China Grant 2015CB358700, the National Natural Science Foundation of China Grant 61822203 ,61772297, 61632016, 61761146003, and a grant from Microsoft Research Asia.

## Footnotes

[2]Without loss of generality, we assume all $D$ features are missing at those steps for the sake of clarity.

[3]In this paper, we simply adopt mean pooling. The design of attention mechanism is out of this paper's scope.

[4]https://github.com/caow13/BRITS

[5]https://github.com/iskandr/fancyimpute

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
