[Reviews · NeurIPS 2018]

Reviewer 1



The paper proposed a bidirectional RNN model for imputing missing values in multivariate time series. The proposed model is evaluated on three types of datasets and compared to a range of baseline methods. The experiments show that the proposed model outperforms all baseline methods in terms of both imputation quality and the performance of supervised learning tasks. Overall the paper is quite easy to follow. The second paragraph of the introduction section and the related work section provide a comprehensive list of existing frameworks for missing value imputation in general or methods that are specific to time series, which motivates the development of the proposed work. The paper would be more readable if it is more explicit on the dimensionality of the model parameters such as W_x, W_\gamma, W_h and so on. Although those can be inferred given that the hidden states are 64-dimensional (line 228) and each x_t is D-dimensional, explicitly specifying that, for example, W_x is D by 64, W_h is 64 by 64 and gamma_t is 64-dimensional might emphasize the fact that the correlations between \hat{x}_t are implicitly captured by the shared 64-dimensional hidden states. When D >> 64, we can even view the construction of \hat{x}_t as a (low dimensional) nonlinear factorization according to equations 1 to 4. “Intuitively, if \delta_t is large, we expect a small \gamma_t to decay the hidden state.” (between line 132 and 133): If smaller \gamma is expected for larger \delta, should we put certain constraint over W_\gamma such as positivity? This also leads to the question of whether we should always “decay” the hidden state as the gap delta gets larger. From equation 4, since h_t is quite convolved with h_{t-1} and x_t^c when step t is missing, especially as x_t^c is also a function of h_t, it’s not clear how to justify the decaying of the term associated with h_{t-1} but leaving the x_t^c term intact. Is it possible that the actual dynamic favors focusing more on h_{t-1} when \delta is large? Also, is it necessary to restrict the values of \gamma_t to between zero and one by equation 3? It’d be a great addition to justify the design choice regarding these concerns, either in theory or through some ablative experiments. Moreover, regardless of missing entries, Definition 1 mentioned that the time gap between different timestamps may not be the same. Considering the case that all D features are simultaneously sampled but in a very non-uniform rate (\delta_t varies a lot for different t), how to justify the choice of equation 3 and 4 under this setting? A related work addressing this issue is “Phased LSTM: Accelerating Recurrent Network Training for Long or Event-based Sequences” by Daniel Neil, et al. Two choices of \alpha_i are mentioned in line 134-135. Which one is used in the experiments? Equation 7 explicitly assumes that features are linearly correlated. However, in some real-world applications, this might not be the case. For example, sometimes in the multivariate time series data, one feature might be strongly correlated with the other but with a small time delay. This is commonly seen in the healthcare data especially when it comes to physiological signals. I wonder how would BRITS compare with architectures such as multi-layer RNN with vector outputs, which could potentially capture nonlinear correlation across features but with less ad-hoc components. “The supervision of classification/regression makes the estimation of missing values more accurate.” (line 51-52) It’d be more convincing if showing some results of, for example, comparing BRITS with and without supervision signals. It’s not clear if the mean relative error (MRE) described in line 235-236 is a sensible metric as |pred_i - label_i| is not some normalized quantity such as accuracy and representing it in percentage seems unnecessary, although we can see that it is strongly correlated with MAE in Table 1. I really like the experiments in the paper that compare with a wide range of related methods. It would make the experiments more reproducible if the paper reports (perhaps in the appendix) the hyperparameters used by each of the baseline methods and their architecture in detail. Moreover, reporting something like the number of parameters used in each of the models especially for GRU-D, M-RNN and the proposed models would be quite helpful for interpreting the experimental results. For example, if BRITS has significantly more parameters than BRITS-I due to the extra \hat{z}_t terms (section 4.3), then it is not clear if the advantage of BRITS shown in Table 1 comes from having more parameters, which we expect the model to be more expressive, or comes from the extra “feature-based estimation”. The similar question also applies to the comparison with GRU-D and M-RNN. Minor: Example 1 might be a bit misleading by referring that x_5, x_6, and x_7 are missing, as it somewhat suggests that all D features are missing at those timestamps, but Definition 1 allows different missingness across features. Therefore, L_e(\hat{x}_t, x_t) should be able to handle the case with partial observations. Minor: Figure 3 compares the imputation results by ImputeTS and BRITS-I, and in the text followed claims that BRITS-I demonstrates better performance than ImupteTS. However, it seems that ImputeTS better captures local patterns especially in the case in the middle column. ImputeTS performs poorly mostly at the beginning of the time series. What if a “bidirectional-version” of the ImputeTS is used and how would that compare with BRITS-I? Post-rebuttal comments: I appreciate that the authors have addressed most of my concerns and make things more clear in their response. For the discussion regarding the decay factor, I still have doubt about the fundamental modeling choice here. According to equations 3 and 4, the “decay factor” \gamma_t only applies to h_{t-1} but not the term associated with U_t. In the state space model, we usually assume that the hidden states h capture the actual transition of the dynamics, while the observation x could be noisy due to either sensory error or missingness. This implies that h should be more reliable than x. However, here in equation 4, h is decayed by gamma but not for x. I feel an alternative idea is to decay the U_h term instead but applying some transition function over h_{t-1} accounting for \delta_t. Empirically, it’d be helpful to compare different decaying schemes: decaying the W_h term, decaying the U_h term, or both to justify the modeling choice. A minor extra question I have is about the KNN imputation in Section 5.2.2: how is the similarity defined in the presence of missing data?

Reviewer 2



Post-rebuttal comment: The authors responded to some of my concerns. In particular, they addressed my question about the utility of this model to do imputation independent of the supervision task. According to their response, the imputed data produced by BRITS is superior in terms of downstream classification performance than forward-filled data, demonstrating utility. If this experiment can be extended to include comparisons from the other imputation methods then the paper will be strengthened. On this assumption, I change my score from a 5 to 6. Summary of work: They describe a RNN-based framework for performing imputation and simultaneous classification of time series. The RNNs are trained to predict the next (or previous) value of the time-series , with a consistency requirement between forward and backward RNNs. The architecture makes use of masking variables to denote missing values, and time deltas since the last observed value. The latter are used to decay parts of the hidden state, and to weight the relative contributions to the imputed value from other variables, or from the historical prediction of that variable. Since imputation is performed at the same time as classification, they report both imputation performance and AUROC on three real-world datasets. Explanation of overall score: I think this approach is solid, but the analyses are slightly lacking, limiting the general interest and applicability of the approach. Explanation of confidence score: I am reasonably familiar with the related work, and there isn’t much mathematics to check. Quality: * The proposed model and variants are evaluated on three datasets for imputation, and two for imputation+binary classification. * I like the analysis in figure 4/appendix C, this is a useful experiment. * That the “correlated recurrent imputation” helps much over the independent version ((B)RITS v. (B)RITS-I) for imputation is only evident on the healthcare dataset - in the other cases, the difference is of the order of 0.1 percentage points or so (and RITS-I confusingly even outperforms RITS on the Human Activity dataset). Depending on the nature of the missingness this doesn’t seem entirely surprising - the difference between the independent and correlated prediction will depend on: 1) how correlated the variables actually are, and 2) how many non-missing values are observed alongside the data-point to be imputed. It would be interesting to see if these factors could explain the relative performance of BRITS v. BRITS-I on the different datasets. * How is the performance of GRU-D for imputation measured if it doesn’t impute missing data explicitly? * In applied machine learning, imputation is usually done as a precursor step to e.g. a supervised learning problem. I’m glad to see that performance on subsequent tasks is included in this work, but I would like to see more detail here. As indicated by the relatively poor imputation performance of GRU-D (although this may be related to the fact that it doesn’t perform imputation, as per my previous question), but the high classification performance, it would be interesting to see more about how imputation accuracy actually impacts downstream classification. I think to do this in a fair way you would need to separate the imputation and classification tasks - get each model to first perform imputation on the dataset, and then train the same classifier architecture (e.g. a random forest) on each imputed version of the dataset, and compare performance. This would enable the imputation performed by the non-RNN baselines to be compared. You could then also compare how training the RNN to classify at the same time as impute (as is done already) presumably improves classification performance, and potentially changes the imputed data. My intuition here is that certain imputation strategies may appear poor by absolute error, but may actually be adequate for the types of downstream tasks considered. * Highlighting a sub-part of the previous point - I think it would be interesting independent of the above investigation to see how much of an effect the classification task has on the imputation performance - that is, to turn off L_out in the loss. It would be an interesting result (and perhaps a useful contribution to the ongoing discussion around unsupervised learning) to demonstrate that having a task can improve imputation performance - and what would be doubly interesting would be to show that the resulting imputed data is still useful for *other* tasks! * We see empirically that BRITS outperforms GRU-D and MRNN, and the use of feature correlations is given as an explanation, but (B)RITS-I also outperforms both models, so something else must be going on. Is there something else about the (B)RITS architecture that’s providing an advantage? * One question: is it feasible to run this model to perform imputation in real-time? * In summary, I think the experiments here are interesting, but there are many more potentially interesting analyses which could be run. Clarity: * I like how this paper was laid out - the RITS-I -> BRITS-I -> BRITS progression made it quite easy to see how the model is constructed. * I think it would be useful to highlight where data is missing at random, or not-at-random in these experiments. I think it’s not at random in the Air Quality dataset, but at random in the others? The descriptions (e.g. figure 1, example 1) make it sound like the primary form of missingness is missing in blocks, which is slightly misleading. * Slight inconsistency: in the GRU-D paper, the AUROC on mortality in the PhysioNet data is reported at 0.8424 ± 0.012, which would make it comparable to the performance of BRITS (0.850 ± 0.002), but in this submission it is reported at 0.828 ± 0.004. * Minor: I think that errors should not be neglected when choosing which result to put in boldface (see last two rows, second column in Table 2 - these results are within error of each other). * Regarding reproducibility, I greatly appreciate that the authors provided an anonymous code repository, however the details of how the other RNN models were trained are missing. Originality: * It would be helpful if the other (GRU-D, MRNN) methods were described in a little more detail to make clear which elements of this model are original. Significance: * Solving problems on time-series data with missing data is quite important, so this work is significant in that sense. * However, I feel the analyses are slightly limited, making it hard to conclude what exactly about this architecture is providing benefit, or how generally useful it is for imputation (for example, in cases where the classification problem is not known ahead of time, or the classification architecture will not also be an RNN).

Reviewer 3



The authors present a method to handle missing values in sequential data by training an RNN to predict the value of the next variable in a sequence and use the predictions instead of the ground truth when missing. They validate their approach on three datasets with different amounts of (naturally or artificially) missing data where they outperform relevant baselines. The method is mostly clearly presented and easy to follow, although some justifications are missing or not quite to the point. Why does \mathcal{L}_e use an l_1 distance and l_t^{cons} an l_2 loss, even though they seem to serve similar purposes? Also, the justification for Equation (7) is the lack of correlations between features of \hat{x}_t in Equation (1); but these can correlated, since they are obtained from a common h_{t-1}. So does the added layer really need to have zeroed out diagonal elements? The experiments section validates the author's claims, but is missing important details. The accuracy results for prediction really need to be in the main body, even if something else (e.g. the definition of MAE or MRE) has to be moved to the appendix. (Che et al., 2018) report an AUC of 0.8424 in their paper, why is it 0.828 here? Finally, as mentioned above, the importance of having a zero diagonal in W_z needs to be experimentally validated.